# OpenReview forum: "AutoCustomization: A Unified Framework for Effortless, Selective LLM Bias and Style Finetuning"
_ICLR.cc/2025/Conference — Submitted to ICLR 2025_

### Official Review · Reviewer_H32M · 2024-11-03

**Soundness:** 1
**Presentation:** 2
**Contribution:** 1
**Rating:** 3
**Confidence:** 4

**Summary:**

This paper proposes an approach (AutoCustomization) to customize LLMs to a predefined property, such as Republican (vs. Democrat). It uses an LLM to first generate subareas representing the differences between two opposite properties, and then for each subarea generate questions and their corresponding answer pairs representing the two opposite properties. Next, it uses aforementioned synthetic data to fine-tune an LLM with the objective of increasing the generation probability of the desired property while decreasing the generation probability of the opposite property.

Evaluation results show that AutoCustomization is more robust than a prompting method to random distracting context and adversarial user prompts. It also shows that a proposed metric for bias adjustment (BiasShift) correlates better than LLM-based evaluators to human labels. It even aligns with human labels better than human evaluations between themselves.

**Strengths:**

- This paper successfully applies the framework of synthetic data generation + fine-tuning to the application of polar property customization.
- It provides data points showing that a fine-tuned model is more robust than a prompting method to random distracting context and adversarial user prompts.

**Weaknesses:**

- There are no convincing evidences showing that the proposed method is better then prompting approaches. (1) Based on the results in Figure 4 (left) with zero prompt padding, the prompting method is better than the proposed method on regular queries. (2) The baseline prompt is too simple, and this paper does not introduce prompting-based related work to compare with. (3) In terms of robustness to random distracting context and adversarial user prompts, some prompt engineering could significantly mitigate the issue, such as putting the system prompt after the user prompt.
- There are no in-depth discussions or analysis on the weakness of the proposed approach. For example, fine-tuning on a small and specific dataset could lead to catastrophic forgetting. This paper augments selected areas of the MMLU dataset, but with no detailed analysis on its impact. Is it a generalizable solution?
- The claim that BiasShift is an evaluation metric that is better than other LLM-based evaluators and human is misleading. (1) It is a training objective, not a proper evaluation metric. It "cannot be utilized to compare the bias shift for different models". How can an evaluation metric useful if it cannot be used to compare different models? (2) BiasShift is used on identically distributed training and test data, but LLM-based evaluators are few-shot. It's not a fair comparison. (3) There is no analysis on why human annotators don't agree. It could be due to unqualified annotators, ambiguous examples, etc. To claim "super-human", we need to first define the ground-truth.
- This paper uses a widely adapted synthetic data generation + fine-tuning framework without significant scientific contributions to the broader ICLR community.

**Questions:**

- Figure 2. "By calculating the embedding cosine similarity for the continuations in the given set of subareas and their continuations." This sentence is hard to understand. How to calculate subarea similarity w.r.t. continuations?
- Line 193. incomplete sentence
- Section 4.1.1. How many human annotators an example is labeled by? For each example, what is the final label used to compare with automatic metrics?

---

### Official Review · Reviewer_qADc · 2024-11-03

**Soundness:** 1
**Presentation:** 2
**Contribution:** 1
**Rating:** 1
**Confidence:** 4

**Summary:**

The paper proposes to use LLM to generate  fine-tuning datasets focused on a specific customization axis on bias or style for model fine-tuning.
The paper claims that compared to prompting method, this method has better performance, is more effortless and more robust.

**Strengths:**

The paper is relatively well written.

**Weaknesses:**

The experiments don’t support the paper’s claim.

Throughout the experiments, this paper lacks thorough comparison with their only category of baseline methods, prompting based methods. If the paper wants to claim that their method is superior to prompting methods, then it has to thoroughly try different prompting methods, like few-shot, chain of thought, multi-agent debate, reflection, etc, and to show that the proposed method indeed presents superior performance compared to such prompt methods. None of the experiments presented in the paper directly shows that. Moreover, in figure 4 (a), it seems that prompting method outperforms the proposed method with no pending.

Similarly, the paper claims that  prompting methods are “often cumbersome and brittle, requiring complex techniques and prompt libraries tailored to specific models and tasks”. I don’t understand what this means. I think intuitively fine-tuning a model is more “cumbersome”. There are just so many such arbitrary and subjective claims that are not supported well by exps or in this case, I cannot think of an exp design backing this claim up.

**Questions:**

Please see weakness

---

### Official Review · Reviewer_u7Sp · 2024-11-05

**Soundness:** 1
**Presentation:** 2
**Contribution:** 2
**Rating:** 3
**Confidence:** 4

**Summary:**

In this paper, the authors propose the AutoCustomization method to control a model's tendency to prefer one tendency over another along an "axis of adjustment" such as political bias between Republicans and Democrats. In this method, they use an LLM to generate a dataset of relevant queries to a topic, and a model response corresponding to each pole of that axis for each query. They then use fine-tuning to make the model prefer either one pole or another. They propose an evaluation metric called BiasShift to judge the effectiveness of this method in increasing/decreasing bias toward a pole and empirically compare AutoCustomization's BiasShift to human annotators.

**Strengths:**

Originality: to my knowledge, this exact method has not been proposed by others before. The method wires together LLM-based data generation and fine-tuning to the end of making a model more amenable to one behavior or another.

Quality: the paper presents one case study bias axis (Republican vs Democrat) and shows that the BiasShift value produced by their method on a range of examples correlates well with human-annotated bias rankings of those examples.

Clarity: the main method/algorithm 1 is described clearly and it is easy to grasp the exact approach taken by the authors. I believe it would be straightforward to reproduce from the description in the paper.

Significance: Aligning models to behave according to user-specified intentions is important, so developing a method in service of this is significant.

**Weaknesses:**

The primary issue I see with this paper is a lack of empirical evidence to support the claims in the paper.

Some of the claims seem rather strong, i.e., BiasShift "proves to have super-human reliability". If I understand correctly, its reliability being "super-human" is judged by its degree of ranking correlation with human annotations compared to the average human annotation. I am not convinced that having a higher ranking correlation than the average would classify the method to be at a "superhuman" reliability. One further concern about using ranking is that ranking with 150 elements in the list might make the human annotation job especially difficult and under-estimate human ability to judge bias. While it is more difficult for a human to rank 150 examples reliably, it might be more apt to evaluate both human and model bias estimates based on pairwise comparisons.

Second (unless I'm missing this somewhere), the quantitative evaluations in the paper on the BiasShift of this method appears to be only applied to the Republican vs Democrat example (Section 4.1). While around 12 different bias axes are mentioned in other parts of the paper, those other axes do not seem to have the same quantitative evaluations included in the paper. The efficacy of this method needs to be shown on many different axes.

Third, the claim that the method is computationally cheap sounds rather subjective. Indeed, compared to prompting, it would be arguably more expensive (given that this method requires both data generation + fine-tuning). Is there another method to compare to regarding computational efficiency?

Finally, there's some possible improvements to clarity that could help the reader better understand the experiments and analysis to support the paper's main claims (listed below in the questions section).

Overall, in my view, substantive improvements (primarily in expanding the experimentation + taking a little more care in the claims) need to be made. However, I think this draft has promise for being useful to the alignment/interpretability community and, if these concerns are alleviated, could be a meaningful contribution (either at this conference or in a future one).

**Questions:**

* Is BiasShift something that can be inherently interpretable in some way?
* In Table 1, how is the bias determined for the different models listed?
* In Figure 4, how is the manifested bias computed/measured?
* Can BiasShift also be disentangled from the AutoCustomization fine-tuning method and computed independently for, e.g., a model prompted to follow one ideology and not prompted at all? This seems like a useful baseline for understanding the effectiveness of AutoCustomization vs prompting.

---

### Official Review · Reviewer_26bK · 2024-11-05

**Soundness:** 2
**Presentation:** 2
**Contribution:** 2
**Rating:** 3
**Confidence:** 4

**Summary:**

This paper presents a framework that prompts LLMs to generate data that shows the opposite biases on a specified bias axis. Experiments show that the proposed framework outperforms the traditional prompting techniques and provides a more reliable way of adjusting the bias in model responses.

**Strengths:**

This paper introduces a framework designed to prompt an LLM to generate data that exhibits biases opposite to those specified along a given bias axis, and then finetune the LLM on the generated data using an introduced metric called BiasShift. By fine-tuning on such paired-data, the model is expected to have a higher chance of generating text with the target bias.

**Weaknesses:**

* The presentation is bad, e.g. Figure 2 is illegible.
* This paper lacks details (e.g.,  the number of annotators, annotator profiles that might influence their rankings) about the important human evaluation in Section 4.1. Also, some information is confusing, e.g., how were the 150 answers generated? Were they answers to the same question or all the questions?
* The dataset generation method described in Section 3.1 is essentially applying the approach proposed in the prior work (https://arxiv.org/abs/2306.15895), which is incremental.
* Some conclusions drawn from the experiments seem overclaimed, and they might need more experiments and analyses. For example, Figure 4 shows that AutoCustomization can adapt bias more robustly compared to PromptEngineering. However, the compared baseline "PromptEngineering" is too naive and weak, employing a single and unoptimized prompt. This setup potentially skews the comparison, making the AutoCustomization appear more effective than it might be against a more sophisticated baseline.

**Questions:**

* How is the "strong, domain-specific classifier" trained which is used for bias grading in L345? What is its performance? What are the 200 examples used for training this classifier?
* What LLMs are used for data generation in Section 3.1? What is the target LLM in Section 3.2?

---

### Official Review · Reviewer_ynyL · 2024-11-11

**Soundness:** 3
**Presentation:** 3
**Contribution:** 1
**Rating:** 3
**Confidence:** 4

**Summary:**

The work presents AutoCustomization, which aims to adapt a language model's generation style towards one of a two opposite axis of bias (e.g., democrat and republication). The method works by first prompting the language model to create a set of subareas of the bias, and then prompting the model again to synthesize a set of sentences for each subarea for each stance. Training is then carried out by maximizing the probability of generating the sentences from the desired stance and minimizing the probability of generating the sentences from the undesired one, while being regularized by maintaining the capability on a neutral dataset, MMLU in this case. The training is also monitored by a biasShift value, which the author use as early stopping. The experiments carried out on several different axes where the author compared with prompting the language model as an adaptation.

**Strengths:**

The paper is written clearly, with the method simple and easy to understand.

**Weaknesses:**

The paper seems to have ignored a large chunk of related work on supervision-less adaption to desired styles. Much of them have similarities. For example, [1, 2]. Thus, the contribution of data-creation in the work seems to be limited.
The training method also seems to have high relationships with the large community of preference learning, DPO, Slic, among many others. The author did not mention, nor compare, these in this work, making it hard to assess the contribution of the training method proposed.
Combined, the author did not compare with a reasonable baseline in the experimental part.


[1] Instruction Tuning with GPT-4
[2] Learning Preference Model for LLMs via Automatic Preference Data Generation

**Questions:**

1. t in BiasShift seems undefined.

---

### Meta-Review · Area_Chair_WPSR · 2024-12-14

**Metareview:**

The paper proposes a method that generates data of an opposing bias.
Strengths:
new framework
Weaknesses:
The paper lacks in details, misses related work, is not clear enough and overclaims.

**Additional Comments On Reviewer Discussion:**

NA

---

### Decision · Program_Chairs · 2025-01-22

Reject